Taxonomic relevance of petiole anatomical and micro-morphological characteristics of Clematis L. (Ranunculaceae) taxa from South Korea

Park Beom Kyun
Son Dong Chan
Ghimire Balkrishna ghimre2ab@gmail.com
Division of Forest Biodiversity, Korea National Arboretum , Pocheon , South Korea
Sosa Victoria
Electronic publication date: 2021 Jun 29
Publication date: 2021
Volume: 9
Electronic Location ID: e11669
Received 2020 Dec 9; Accepted 2021 Jun 3
Copyright: ©2021 Park et al.
Copyright year: 2021
Copyright holder: Park et al.
License: This is an open access article distributed under the terms of the Creative Commons Attribution License, which permits unrestricted use, distribution, reproduction and adaptation in any medium and for any purpose provided that it is properly attributed. For attribution, the original author(s), title, publication source (PeerJ) and either DOI or URL of the article must be cited.
License URL: https://creativecommons.org/licenses/by/4.0/

Keywords: Petiole morphology, Clematis, Trichomes, Vascular bundle, Infrageneric relationship, Taxonomy

Funding: Silvics of Korea (KNA1–1–18, 15–3) This study including the APC was financially supported by the project “Silvics of Korea (KNA1–1–18, 15–3)”. The funders had no role in study design, data collection and analysis, decision to publish, or preparation of the manuscript.

==============================
We assessed the micro-morphological and anatomical structures of the petioles of 19 Clematis taxa from South Korea. The petiole surface features were observed with the help of stereomicroscopy and scanning electron microscopy, and the anatomical features are studied via microtomy and light microscopy. The results of this study showed that the presence/absence and abundance of trichomes, petiole cross-section outlines, upper surface wings and grooves, and the number of vascular bundles were useful for species discrimination in Clematis. Among the studied taxa, C. hexapetala was the only species with a glabrous petiole surface. Two types of trichomes were observed in the other 18 taxa: long, non-glandular and flagelliform trichomes and short, glandular capitate trichomes. We found four to six major vascular bundles and a maximum of eight interfascicular vascular bundles (C. heracleifolia and C. urticifolia) in the 19 taxa. A cluster analysis based on UPGMA identified six clusters with 18 nodes. Although the number of taxa investigated was limited, taxa from the sections Tubulosae, Viorna, and Astragene clustered with each other in the UPGMA phenogram due to the overall similarity of petiole features. Based on this observation, we can conclude that most of the petiole features are limited to the species level, and thus, the data obtained could be used as descriptive and/or diagnostic features for particular taxa, which may be useful for the investigation of problematic taxa in the genus.

Introduction

Ranunculaceae is one of the larger families of eudicots, comprising nearly 2,500 species within 50–60 genera (Tamura, 1993; Hoot, Meyer & Manning, 2012; Wang et al., 2013). The family has been classified under the early branching order of eudicot Ranunculales (The Angiosperm Phylogeny Group, 2016). Ranunculaceae are distributed across the world and exhibit their greatest diversity in the temperate and cold regions of the Northern and Southern Hemispheres (Tamura, 1993). A number of classification models that consider morphological characters, molecular sequencing, and a combinations of both morphological and molecular data have been proposed for this family (Hutchinson, 1923; Janchen, 1949; Johansson & Jansen, 1993; Tamura, 1995; Hoot, 1995; Jensen et al., 1995; Ro, Keener & McPheron, 1997; Wang et al., 2009; Wang et al., 2013; Emadzade et al., 2010; Hoot, Meyer & Manning, 2012; Wang et al., 2014; Zhai et al., 2019). In addition to the various morphological criteria, the basic chromosome numbers and types of chromosomes have become reliable references that are compatible with the molecular phylogeny of Ranunculaceae (Gregory, 1941; Tamura, 1987; Ro, Keener & McPheron, 1997; Wang et al., 2009; Heywood et al., 2007).

Within Ranunculaceae, Clematis L. is classified under the tribe Anemoneae DC. in the subfamily Ranunculoideae Hutch. (Tamura, 1995). Clematis is one of the largest genera in the family, which comprises approximately 280–350 cosmopolitan species (Tamura, 1987; Tamura, 1995; Wang & Li, 2005). In Korea, Nakai (1952) reported 21 species and 14 varieties of Clematis in a synoptical sketch of Korean flora, but Lee (1967) later identified 16 species, 11 varieties, and five forma in the genus. In the book New Flora of Korea, Lee (2007) described 18 taxa, including C. taeguensis Y. Lee, which was first described by Lee (1982). The Korea National Arboretum (2017) recently listed 17 species and five varieties of Clematis in the Checklist of Vascular Plants in Korea, whereas Kim (2017) described 13 species and seven varieties within the genus in The Flora of Korea. After a careful review of Lee (2007); Chang, Kim & Chang (2011), the Korea National Arboretum (2017), and Kim (2017),we included 16 species and three varieties in this study.

Due to the vast morphological disparity among its species Clematis has been a subject of investigation since the early 19th century and has been subjected to several infrageneric revisions. Several systematic studies based on the anatomy of different parts, palynology, and cytology of Clematis have been carried out (Tobe, 1974; Tobe, 1980a; Tobe, 1980b; Tobe, 1980c; Tobe, 1980d; Essig, 1991; Zhang & He, 1991; Yano, 1993; Yang & Moore, 1999; Shi & Li, 2003; Xie & Li, 2012; Ghimire et al., 2020). The morphological characters that have been extensively studied and considered in the infrageneric classifications of Clematis include the habit, seed germination, seedling phyllotaxy, leaf structure, inflorescence vertical structure, floral morphology, and pollen and achene morphology (see Wang & Li, 2005).

The importance of the nodal and petiolar anatomy in intergeneric and familial taxonomy has been extensively studied (Howard, 1962; Howard, 1979; Schofield, 1968; Dickison, 1969; Dickison, 1980; Datta & Dasgupta, 1979). The middle portion of the petiole is considered the most stable zone, from which even a single section can be taken for comparative purposes (Metcalfe & Chalk, 1979). In addition, the complex vascular systems of the petiole provide a range of diagnostic structures that can be useful for taxonomic treatment at any rank (Solereder, 1908; Metcalfe & Chalk, 1979; Ashton, 1982; Dehgan, 1982; Rojo, 1987; Pedri, Hussin & Latiff, 1991; Kamel & Loufty, 2001; Kocsis & Borhidi, 2003; Talip et al., 2016; Talip et al., 2017). Within Ranunculaceae, few taxonomic-related investigations based on the anatomy of petioles, nodes, and stems have been carried out (Worsdell, 1908; Tamura, 1962; Oh, 1971; Kavathekár & Pillai, 1976; Tobe, 1979; Tobe, 1980a; Tobe, 1980b; Tobe, 1980c; Tobe, 1980d; Kökdil et al., 2006; Gostin, 2011; Novikoff & Mitka, 2015). Unfortunately, studies pertaining to the petiole morphology and anatomy of Clematis, as one of the most morphologically diverse and taxonomically complicated genera within Ranunculaceae, are very rare in the literature. In an anatomical study of Clematis in Korea, only Oh (1971) provided remarks on the petiole anatomy of nine Korean species.

In this study, we provide a comprehensive investigation of the petiole micromorphology and anatomy of 19 Clematis taxa distributed in Korea. The primary objective of this study was to investigate the petiole micromorphological and anatomical structure of the included taxa in detail and to evaluate the implications of petiolar characters for species delimitation. We attempted to compare our results with those obtained for previously studied species and summarized them to reach taxonomic conclusions.

Materials and Methods

Plant materials

The names of the investigated species and their voucher numbers are provided in Table 1. Formal identification of the plant taxa was carried out by a group of plant taxonomists, including Dr. Dong Chan Son (one of the authors) in the Korea National Arboretum. The voucher specimens were deposited in the herbarium of the Korea National Arboretum (KH). Data were collected as previously described in Ghimire et al. (2020)

Table 1 Name of taxa with voucher number and collection information with different classifications.

Lehtonen, Christenhusz & Falck (2016)	Tamura (1987)	Johnson (1997)	Wang & Li (2005)	Taxon	Locality	Voucher No.	
Clade C	Sect. Clematis	Sect. Clematis	Sect. Clematis	C. apiifolia DC.	Mt. Sinbul, Icheon-ri, Sangbuk-myeon, Ulju-gun, Ulsan, Korea	Sinbulsan-190911-001	
Clade C	Sect. Clematis	Sect. Clematis	Sect. Clematis	C. brevicaudata DC.	Unchi-ri, Sindong-eup, Jeongseon-gun, Gangwon-do, Korea	Unchiri-191007-001	
Clade C	Sect. Clematis	Sect. Clematis	Sect. Clematis	C. trichotoma Nakai	Mt. Sinbul, Icheon-ri, Sangbuk-myeon, Ulju-gun, Ulsan, Korea	Sinbulsan-190911-001	
Clade K	Sect. Flammula	Sect. Flammula	Sect. Clematis	C. taeguensis Y. Lee	Gyuram-ri, Jeongseon-eup, Jeongseon-gun, Gangwon-do, Korea	Gyuramri-190818-001	
Clade K	Sect. Angustifolia	Sect. Flammula	Sect. Clematis	C. hexapetala Pall.	Ho-ri, Palbong-myeon, Seosan-si, Chungcheongnam-do, Korea	Hori-190809-001	
Clade K	Sect. Flammula	Sect. Flammula	Sect. Clematis	C. terniflora DC.	Jukpo-ri, Dolsan-eup, Yeosu-si, Jeollanam-do, Korea	Dolsando-191004-002	
Clade K	Sect. Flammula	Sect. Flammula	Sect. Clematis	C. terniflora var. mandshurica (Rupr.) Ohwi	Namhansanseong Fortress, Sanseong-ri, Namhansanseong-myeon, Gwangju-si, Gyeonggi-do, Korea	Namhansanseong-190809-001	
Clade C	Sect. Tubulosae	Sect. Tubulosae	Sect. Tubulosae	C. heracleifolia DC.	Sihwa Lake, Munho-ri, Namyang-eup, Hwaseong-si, Gyeonggi-do, Korea	Sihwaho-190921-016	
Clade C	Sect. Tubulosae	Sect. Tubulosae	Sect. Tubulosae	C. urticifolia Nakai ex Kitag.	Mt. Gariwang, Sugam-ri, Bukpyeong-myeon, Jeongseon-gun, Gangwon-do, Korea	Gariwangsan-191007-001	
Clade C	Sect. Tubulosae	Sect. Tubulosae	Sect. Tubulosae	C. takedana Makino	Sihwa Lake, Munho-ri, Namyang-eup, Hwaseong-si, Gyeonggi-do, Korea	Sihwaho-190921-001	
Clade L	Sect. Viticella	Sect. Viticella	Sect. Viticella	C. patens C.Morren & Dence.	Mt. Johang, Samsong-ri, Cheongcheon-myeon, Goesan-gun, Chungcheongbuk-do, Korea	Johangsan-170831-049	
Clade K	Sect. Pterocarpa	Sect. Pterocarpa	Sect. Pterocarpa	C. brachyura Maxim.	Seondol, Bangjeol-ri, Yeongwol-eup, Yeongwol-gun, Gangwon-do, Korea	Seondol-190719-001	
Clade I	Sect. Meclatis	Sect. Meclatis	Sect. Meclatis	C. serratifolia Rehder	Gasong-ri, Dosan-myeon, Andong-si, Gyeongsangbuk-do, Korea	Gasongri-191007-001	
Clade L	Sect. Viorna	Sect. Viorna	Sect. Viorna	C. fusca Turcz.	Mt. Cheongtae, Sapgyo-ri, Dunnae-myeon, Hoengseong-gun, Gangwon-do, Korea	Cheongtaesan-190819-001	
Clade L	Sect. Viorna	Sect. Viorna	Sect. Viorna	C. fusca var. flabellata (Nakai) J. S. Kim	Eundae-bong, Gohan-ri, Gohan-eup, Jeongseon-gun, Gangwon-do, Korea	Eundaebong-190818-001	
Clade L	Sect. Viorna	Sect. Viorna	Sect. Viorna	C. fusca var. violacea Maxim.	Mt. Baekhwa, Mawon-ri, Mungyeong-eup, Mungyeong-si, Gyeongsangbuk-do, Korea	Mungyeongsi(Mawonri, Baekhwasan-150707-007	
Clade H	Sect. Atragene	Sect. Atragene	Sect. Atragene	C. calcicola J. S. Kim	Mt. Deokhang, Daei-ri, Singi-myeon, Samcheok-si, Gangwon-do, Korea	Deokhangsan-190818-001	
Clade H	Sect. Atragene	Sect. Atragene	Sect. Atragene	C. koreana Kom.	Mt. Hambaek, Gohan-ri, Gohan-eup, Jeongseon-gun, Gangwon-do, Korea	Hambaeksan-190818-001	
Clade H	Sect. Atragene	Sect. Atragene	Sect. Atragene	C. ochotensis (Pall.) Poiret	Mt. Gariwang, Sugam-ri, Bukpyeong-myeon, Jeongseon-gun, Gangwon-do, Korea	Gariwangsan-190819-007	

Stereo and scanning electron microscopy

The petiole morphology, including the indumentum, trichome type and abundance, and upper surface groove was observed under a stereomicroscope and a scanning electron microscope (SEM). A Leica MZ16 FA microscope (Leica Microsystems GmbH, Wetzlar, Germany) was used for the observations and digital images of the best-represented part of the petiole were taken with a Leica DFC420 C multifocal camera attached to the microscope. Before SEM imaging, petiole pieces were immersed in 100% ethanol and sputter-coated with gold in a KIC-IA COXEM Ion-Coater (COXEM. Co., Ltd., Daejeon, Korea). SEM imaging was carried out with a COXEM EM-30 PLUS+ table scanning electron microscope (COXEM) at 20 kV at the seed testing laboratory of the Korea National Arboretum.

Microtome and light microscopy

At least three petioles of each taxon were subjected to microtome sectioning according to the following procedure used in Ghimire et al. (2020). Freshly collected leaf petioles were fixed in formalin, acetic acid, and 50% ethyl alcohol (FAA) at a ratio of 5:5:90 for one week and preserved in 50% ethyl alcohol. During the experiment, the preserved petioles were cut into small pieces (approximately two mm) and dehydrated with an ethanol series (50, 70, 80, 90, 95 and 100%). After complete dehydration, the petiole pieces were infiltrated with ethanol/Technovit mixtures (3:1, 1:1, 1:3, and 100% Technovit) and then embedded in Technovit 7100 resin. The embedded materials were cut into serial sections of 4–6 µm thickness using a Leica RM2255 rotary microtome (Leica Microsystems GmbH, Wetzlar, Germany) with disposable blades and attached to a glass slide. The slides were dried using an electric slide warmer for 12 h. The dried slides were stained with 0.1% toluidine blue ‘O’ for 60–90 s, rinsed with water and dried again with the slide warmer for at least 6 h to remove any remaining water (Johansen, 1940). The stained slides were then mounted with Entellan (Merck Co., Darmstadt, Germany) and later examined under a Leica DM3000 LED (Leica Microsystem, Wetzlar, Germany). Photomicrographs were taken with a scientific CMOS camera. Multiple image alignment was performed using Photoshop CS for Windows 2010.

Morphometry and data analysis

Thirteen quantitative characters were categorized and coded binary and/or multistate. The character states and their codes are provided in Supplementary File S1. Principal component analysis (PCA) and cluster analysis using the unweighted pair group (UPGMA) clustering method using the Gower general similarity coefficient were carried out with MultiVariate Statistical Package 3.1 software (MVSP Version 3.1) (Kovach, 1999).

Results

The morphological and anatomical characters of Clematis petioles observed in this study include the petiole indumentum, trichome type and abundance, petiole outline in cross-section (CS), upper surface wings and groove of the petiole, sclerenchyma region, and vascular bundles. All the characters are summarized in Table 2. Selected images of the petioles are provided in Figs. 1–6. The morphological and anatomical features of the petiole are comprehensively described below.

Petiole surface and trichomes

The petiole surface of the studied species is pubescent except in Clematis hexapetala, which has an almost glabrous surface (a few trichomes occur in the region from which leaflets arise) (Table 2, Figs. 1A–1S). Clematis taeguensis and C. serratifolia have subglabrous petiole surfaces, and only a few trichomes occur on the petioles of these species. Two types of trichomes are observed on the petiole: long, non-glandular, flagelliform trichomes and short, glandular, capitate trichomes (Figs. 1A–1S, 2A–2J). Glandular trichomes are usually distributed in the upper surface groove. In some species such as C. terniflora, C. terniflora var. mandshurica, C. brachyura, C. fusca var. fusca, and C. fusca var. violacea, non-glandular trichomes are concentrated only on the upper surface groove. The pubescent species can be categorized as ‘villous’, which are covered with long, soft, and dense hairs (i.e., C. apiifolia, C. brevicaudata, C. heracleifolia, C. urticifolia, and C. takedana), or ‘pilose’, which are covered with soft, weak, thin, and separated hairs, as found in the rest of the species (i.e., C. trichotoma, C. terniflora, C. terniflora var. mandshurica, C. patens, C. brachyuran, C. fusca var. fusca, C. fusca var. flabellata,C. fusca var. violacea, C. koreana, and C. ochotensis). The non-glandular trichomes are either unicate (C. apiifolia, C. brevicaudata, C. heracleifolia, C. urticifolia, and C. takedana) or flabelliform (all other species except C. hexapetala). Based on the trichome density per unit area, the trichome abundance was categorized as high, medium, or low.

Table 2 Morphological and anatomical features of petiole of Clematis species.

Taxon	Petiole surface	Non-glandular trichomes	Glandular trichomes	Trichome abundance	Petiole outline in cross section	Upper surface wings	
C. apiifolia	Villous	Unicate	Present	High	Pentagonal	Inconspicuous/ conspicuous	
C. brevicaudata	Villous	Unicate	Present	Medium	U-shaped	Inconspicuous	
C. trichotoma	Pilose	Flagelliform	Present	Medium	U-shaped	Inconspicuous	
C. taeguensis	Subglabrous/pilose	Flagelliform	Present	Low	Pentagonal	Conspicuous	
C. hexapetala	Glabrous	Absent	Absent	None	U-shaped/ pentagonal	Conspicuous	
C. terniflora	Pilose	Flagelliform	Present	Low	Semi-circular/ U-shaped	Inconspicuous/ conspicuous	
C. terniflora var. mandshurica	Pilose	Flagelliform	Present	Low	Pentagonal	Conspicuous	
C. urticifolia	Villous	Unicate	Present	High	Pentagonal	Conspicuous	
C. heracleifolia	Villous	Unicate	Present	High	U-shaped	Inconspicuous	
C. takedana	Villous	Unicate	Present	High	Pentagonal	Conspicuous	
C. patens	Pilose	Flagelliform	Present	Medium	U-shaped	Inconspicuous	
C. brachyura	Pilose	Flagelliform	Present	Medium (in upper surface groove)	Pentagonal	Conspicuous	
C. serratifolia	Subglabrous/pilose	Flagelliform	Absent	Low	U-shaped/ pentagonal	Conspicuous	
C. fusca var. fusca	Pilose	Flagelliform	Present	Low	Pentagonal	Conspicuous	
C. fusca var. flabellata	Pilose	Flagelliform	Present	Medium	U-shaped	Conspicuous	
C. fusca var. violacea	Pilose	Flagelliform (in groove)	Present	Low	U-shaped	Conspicuous	
C. calcicola	Subglabrous/pilose	Flagelliform	Absent	Low	U-shaped	Inconspicuous/ conspicuous	
C. koreana	Pilose	Flagelliform	Present	Medium	U-shaped	Conspicuous	
C. ochotensis	Pilose	Flagelliform	Present	Medium	U-shaped	Conspicuous	
Taxon	Upper surface groove	Phloem fiber cap	Interfascicular sclerenchyma	MVB	IVB	TVB	BVG	
C. apiifolia	Flattened/Sub-flattened	Medium, 5-10 layers	<5 layers	6	0	6	1	
C. brevicaudata	Sub flattened	Large, >10 layers	>10 layers	5	2 to 3	7 to 8	2	
C. trichotoma	Sub flattened	Large, >10 layers	5-10 layers	5	2	7	2	
C. taeguensis	Sub flattened	Large, >10 layers	>10 layers	6	4	10	1	
C. hexapetala	V-shaped	Medium, 5-10 layers	<5 layers	5	0	5	0	
C. terniflora	Flattened/Sub flattened	Small, <5 layers	<5 layers	6	0	6	1	
C. terniflora var. mandshurica	Sub flattened	Large, >10 layers	>10 layers	6	4	10	1	
C. urticifolia	Sub flattened	Large, >10 layers	5-10 layers	5	8	13	4	
C. heracleifolia	Sub flattened	Large, >10 layers	<5 layers	6	8	14	3	
C. takedana	Sub flattened	Medium, 5-10 layers	5-10 layers	6	7	13	3	
C. patens	Flattened	Medium, 5-10 layers	>10 layers	4	2	6	1	
C. brachyura	V-shaped	Medium, 5-10 layers	<5 layers	5	4	9	2	
C. serratifolia	Sub flattened/U-shaped	Large, >10 layers	5-10 layers	5	3	8	1	
C. fusca var. fusca	U-shaped	Large, >10 layers	<5 layers	5	2	7	2	
C. fusca var. flabellata	U-shaped	Small, <5 layers	<5 layers	5	5	10	3	
C. fusca var. violacea	Sub flattened	Large, >10 layers	<5 layers	5	2	7	2	
C. calcicola	Sub flattened/U-shaped	Medium, 5-10 layers	<5 layers	5	4	9	2	
C. koreana	V-shaped	Small, <5 layers	<5 layers	5	3	8	2	
C. ochotensis	U-shaped	Medium, 5-10 layers	<5 layers	5	0	5	0	
Notes.

Abbreviations MVB major vascular bundles

IVB interfascicular vascular bundle

TVB total vascular bundle

VBG vascular bundles in groove

Figure 1 Petiole of Clematis under stereomicroscope.

(A) C. apiifolia. (B) C. brevicaudata. (C) C. trichotoma. (D) C. taeguensis. (E) C. hexapetala. (F) C. terniflora. (G) C. terniflora var. mandshurica. (H) C. heraclefolia. (I) C. urticifolia. (J) C. takedana. (K) C. patens. (L) C. brachyura. (M. C. serratifolia. (N) C. fusca var. fusca. (O) C. fusca var. flabellata. (P) C . fusca var. violacea. (Q) C. calcicola. (R) C. koreana. (S) C. ochotensis. Scale bars: one mm.

Figure 2 Scanning electron micrograph of petiole of Clematis.

(A–B). C. heraclefolia. (C–D). C. taeguensis. (E–F). C. patens. (G–H). C. brevicaudata. (I–J). C . fusca var. violacea. Abbreviations: gt, glandular trichome. Scale bar: 200 µm (A, C, E, G, I), 100 µm (B, D, F, H, J).

Petiole outline and upper surface groove

The studied species show considerable variation in their CS petiole outline (Table 2). The species can be divided into three categories based on the shape of their petioles in CS: pentagonal petioles (seven species), U-shaped petioles (nine species), and U-shaped or semi-circular petioles (only C. terniflora) (shown in Figs. 3–6). Clematis hexapetala and C. serratifolia exhibit both pentagonal and semi-circular petioles in CS. Out of the 19 taxa, 13 taxa exhibited two noticeable upper or dorsal surface wings, while six species did not have noticeable wings. Based on the shape of the upper surface groove, the petioles can be categorized as flattened (three species), sub-flattened (11 species), U-shaped (three species), or V-shaped (three species). The formation of the upper surface groove is due primarily to the dorsal surface wings, although some species with inconspicuous wings have a slight groove in the petiole (C. brevicaudata, C. trichotoma, and C. heracleifolia).

Figure 3 Cross section of petiole of Clematis..

(A–B) C. apiifolia. (C–D) C. brevicaudata. (E–F) C. trichotoma. (G–H) C. taeguensis (I–J) C. hexapetala. (K–L) C. terniflora. Abbreviations: co, collenchyma; cu, cuticle; ep, epidermis; ph, phloem; phf, phloem fiber; s, stomata; xy, xylem. Scale bars: 50 µm (B, D, J), 75 µm (F, H, L), 100 µm (A, C, I), 200 µm (E, G, K).

Petiole epidermis and cortex

The epidermis is single-layered and cutinized in all species (Figs. 3–6). The cells are small, rounded, cuboidal, narrow, or slightly elongated. In some places, the continuation of the epidermis is interrupted by the presence of stomata. Stomata are important regulators of gas and water exchange in plants and are usually found in the leaves. They can be found in the stem and petiole but are less prominent in comparison to the leaf. The epidermis is underlain by the cortex, which is 3–5 cells thick. The cortical cells are loosely arranged and parenchymatous with abundant air spaces and are rounded, ovoid, elongated, or irregular in shape. The cortex is collenchymatous above the phloem patches, where the cells are thick-walled and closely packed.

Vascular bundles

The vascular bundles are of the open, conjoint, collateral type. There is remarkable variation in the number of vascular bundles (ranging from five to 14) among Clematis species. The 19 taxa have four to six major vascular bundles and a maximum of eight interfascicular vascular bundles (C. heracleifolia and C. urticifolia). Twelve species have five major vascular bundles, six species have six, and only C. patens has four (Table 2). The major vascular bundles are ovoid with the phloem oriented towards the cortex and the xylem oriented towards the pith. The xylem and phloem are separated by 2-4 layers of cambial cells. Each major vascular bundle is overlain with a cluster of thick-walled fibrous cells, i.e., the phloem fibre cap. The quantity of phloem fibres within the studied species is variable. Based on its height in cell layers, the phloem cap is categorized as large (more than ten cells high), medium (five to ten cells high), or small (less than five cells high). Three species C. terniflora, C. fusca var. fusca, and C. koreana, have small fibre caps, while seven and nine species have medium and large fibre caps, respectively. There is a permanent sclerenchymatous strand between the adjacent vascular bundles in all species.

Figure 4 Cross section of petiole of Clematis..

(A–B). C. terniflora var. mandshurica. (C–D) C. urticifolia. (E–F) C. heraclefolia. (G–H) C. takedana. (I–J) C. patens. (K–L) C. brachyura. Abbreviations: co, collenchyma; cu, cuticle; ep, epidermis; ph, phloem; phf, phloem fiber; s, stomata; sc, sclerenchyma; xy, xylem. Scale bars: Scale bars: 50 µm (B, J, L), 100 µm (A, I, K, F, D, H), 500 µm (C, E, G).

Figure 5 Cross section of petiole of Clematis..

(A–B). C. serratifolia. (C–D) C. fusca var. fusca. (E–F) C. fusca var. flabellata. (G–H) C . fusca var. violacea. (I–J) C. calcicola. (K–L) C. koreana. Abbreviations: co, collenchyma; cu, cuticle; ep, epidermis; ph, phloem; phf, phloem fiber; s, stomata; xy, xylem. Scale bars: 75 µm (B, D, F, H, J, L), 200 µm (A, C, E, G, I, K).

Figure 6 Cross section of petiole of Clematis..

(A–B). C. ochotensis. Abbreviations: co, collenchyma; cu, cuticle; ep, epidermis; ph, phloem; phf, phloem fiber; s, stomata; xy, xylem. Scale bars: 75 µm (B), 200 µm (A).

The petiole of each species has a large region of the ground tissue, i.e., the pith. The cells in the pith are thin-walled, rounded, ovoid, or angular and parenchymatous and are comparatively larger than those in the cortex (Figs. 3–6).

Statistical analysis

The similarities among the species based on the 13 petiole features were revealed using PCA and cluster analysis. The first three components of the PCA explained 74.01% of the total variation in the analysed data. The first axis of the first complete set explained 42.66% of the total variation and showed strong positive loadings for the trichome abundance and the number of vascular bundles (TA, IV, VB, and VG) (Fig. 7). The second axis explained 16.86% of the total variation and showed strong positive loadings for the trichome type and upper surface groove (TT and UG) and strong negative loadings for phloem the fibre cap height and interfascicular sclerenchyma (PF and SC). The cluster analysis based on UPGMA using the Gower similarity coefficient identified six clusters with 18 nodes (Fig. 8). Clematis fusca var. fusca and C. fusca var. violacea, representing the first node in the sixth cluster of the phenogram, showed 93.6% similarity in petiole features, whereas C. hexapetala and C. serratifolia, representing the 18th node in the first cluster in the phenogram, shared only 52.7% similarity in petiole features with the rest of the species.

Figure 7 Principal component analysis (PCA) of 13 petiole characters of Clematis taxa.

PS, petiole surface; TT, trichome type; GT, glandular trichome; TA, trichome abundance; PO, petiole outline in cross-section; UW, upper surface wings; UG, upper surface groove; PF, phloem fiber cap; SC, interfascicular sclerenchyma; MV, major vascular bundles; IV, interfascicular vascular bundle; VB, total vascular bundle; VG, vascular bundles in the groove. Different colour represents different sections of the genus.

Figure 8 UPGMA cluster analysis based on petiole characters of Clematis taxa.

Different colours represents different sections of the genus.

Discussion

The genus Clematis is morphologically diverse in terms of leaf phyllotaxy, types of compound leaves, and leaflet number. The anatomy of the petiole, therefore, is expected to be equally diverse. Previously, Oh (1971) found variation in the number of vascular bundles in the petiole of some Clematis species. Thus, reasonable diversity in petiole anatomical features, including variation in the number of vascular bundles, is certainly useful for the taxonomic treatment of the genus. The overall anatomical organization of the petioles in the investigated species was comparable to that described in Oh (1971). In addition, this study provides a comprehensive anatomical description of the petioles of all Korean Clematis species and a discussion of their taxonomic relevance within the genus, which have not been included in previous studies.

The results of this study showed that Clematis species can be differentiated based on the petiole indumentum. The presence/absence and/or type of trichomes in the petiole have also been used for species differentiation in other taxa (Solereder, 1908; Metcalfe & Chalk, 1979; Talip et al., 2017). Among the 19 species investigated, Clematis hexapetala is the only species with glabrous petioles (although sparsely distributed trichomes can be observed at the base of the leaflets), whereas C. taeguensis and C. serratifolia have subglabrous or sparsely pubescent indumentum. The remaining species have sparsely or thickly pubescent petioles. In sparsely pubescent taxa such as C. terniflora, C. terniflora var. mandshurica, C. fusca var. fusca, and C. fusca var. violacea, the trichomes are restricted to mainly the upper groove region. Additionally, the stem of these taxa is either subglabrous or puberulous only at nodes (Wang & Bartholomew, 2001; Kim, 2017). On the other hand, species such as C. apiifolia, C. heracleifolia, C. urticifolia, and C. takedana, which have thickly pubescent petioles, also have heavily pubescent stems and branches (Wang & Bartholomew, 2001; Kadota, 2006; Moon et al., 2013; Kim, 2017). This indicates that the trichomes in the petiole are generally continuous with those on the stem in the Clematis species.

Most of the species have both types of trichomes, although glandular trichomes are very scarce and restricted to the upper surface groove of the petiole. We observed both glandular capitate and simple non-glandular trichomes on the leaf surface of most of the investigated Clematis species (Ghimire et al. unpublished report). Glandular pubescence on the stem, petiole and leaf of C. gattingeri Small has been reported in a few older studies (Svenson, 1941; Dennis, 1978). Petiole features that are considered to be useful for taxonomic discrimination in various taxa (Solereder, 1908; Metcalfe & Chalk, 1979; Dehgan, 1982; Rojo, 1987; Pedri, Hussin & Latiff, 1991; Kamel & Loufty, 2001; Kocsis & Borhidi, 2003; Talip et al., 2016; Talip et al., 2017) are generally neglected in morphological studies of Clematis because no recent reports have considered the systematic utility of such pubescence for the infra-generic classifications of this genus (Tamura, 1995; Wang & Li, 2005; Lehtonen, Christenhusz & Falck, 2016; Wang & Bartholomew, 2001; Kadota, 2006; Kim, 2017). In a taxonomic study of C. gattingeri, Dennis (1978) confirmed the presence of glandular pubescence on the stem, petiole, and leaf, which had never been observed in other species of subsection Viornae. The recognition of C. gattingeri as a species has been based primarily on the glandular pubescence and small flower size of Gattinger’s specimens. The results from this study also revealed that the petiole indumentum and types of trichomes appear to have taxonomic value for species delimitation in Clematis.

In addition to the surface indumentum, some other petiole features that can contribute to the identification of a particular species in Clematis include the petiole outline in CS, the upper surface groove and wings, and the phloem fibre cap. Of these features, the petiole outline and upper surface groove and wings have already been proven to be useful in the taxonomic discrimination of species in some eudicot genera (Kocsis & Borhidi, 2003; Talip et al., 2017; Abeysinghe & Scharaschkin, 2019). Oh (1971) reported pentagonal and/or horseshoe or rounded horseshoe-shaped petioles in nine Clematis species, and the results of this study corroborated those reports. We observed that the petiole of Clematis in CS is dorsiventral, pentagonal with five visibly and/or weakly represented ridges or semicircular and ridgeless. Clematis hexapetala and C. serratifolia appear to have both pentagonal and U-shaped petioles, whereas some petioles of C. terniflora are semi-circular. Species with pentagonal petioles have conspicuous upper surface wings that form an upper surface groove, while some of the species with U-shaped petioles have inconspicuous upper surface wings. Species with inconspicuous wings such as C. brevicaudata, C. trichotoma, C. heracleifolia, and C. terniflora, still form a slight upper surface groove. Of the three taxa C. fusca var. fusca, C. fusca var. flabellata, and C. fusca var. violacea, the first has a pentagonal petiole, while the latter two have U-shaped petioles with noticeable upper wings. Interestingly, these three taxa showed dissimilar upper surface groove characteristics: C. fusca var. violacea has a sub-flattened groove, whereas C. fusca var. fusca and C. fusca var. flabellata have a U-shaped groove. Although reports on such upper wing extensions and adaxial grooves on the petiole are lacking in the literature, our study suggests the possibility of using these features for species identification in Clematis.

In petiole anatomy, the vascular system of the petiole has received the most attention (Kocsis & Borhidi, 2003; Talip et al., 2016; Talip et al., 2017 Abeysinghe & Scharaschkin, 2019). According to Hare (1942), various arrangements of vascular bundles in the petiole can be used as diagnostic characteristics in some taxonomic groups. Howard (1962), Howard (1974)) suggested that the vascular structure of the petiole is most useful at the generic level and sometimes at the family level, although the intensity of the taxonomic value may vary from one taxonomic group to another. The Clematis petiole showed remarkable consistency in the arrangement of the vascular system, although the studied species differ from each other by the number of major and interfascicular vascular bundles. There are typically five major vascular bundles, which possibly correspond to the five ridges of the petiole; however, C. patens, with exclusively U-shaped petioles, has no ridges, wings, or upper surface grooves, and has only four major vascular bundles. In some taxa, such as C. apiifolia, C. taeguensis, C. hexapetala, C. terniflora var. mandshurica, C. heracleifolia, and C. takedana, the vascular bundle in the upper groove regions develops in the same way as the vascular bundles in the edges, resulting in a total of six major vascular bundles in these taxa.

Regarding the number of vascular bundles, the results of this study are almost congruent with those of Oh (1971) for C. apiifolia, C. trichotoma, and C. koreana but are slightly different for C. brachyura and C. patens, in which he described six major and four interfascicular vascular bundles and six major vascular bundles, respectively. Instead, we observed five major and four interfascicular vascular bundles in C. brachyura and four major and one or two interfascicular bundles in C. patens. We observed variation in the number and position of interfascicular bundles even in the different samples of the same species, including in C. patens, thus, this feature may have only slight taxonomic value for discriminating among Clematis species. On the other hand, the number and position of the major vascular bundles, which showed remarkable consistency among the investigated samples of all species, may have significant taxonomic value for species discrimination within the genus. At this point, our results suggest that the pre-existing data presented by Oh (1971) on the number of vascular bundles specifically, the number of major bundles in the petioles of Clematis species should be corrected.

The cluster analysis based on 13 petiole features generated at least six clusters. Of these, Clematis serratifolia and C. hexapetala formed the first cluster, which was separate from the rest of the species (Fig. 8). The sixth cluster was the largest comprising seven taxa: C. calcicola, C. ochotensis, C. fusca var. fusca, C. fusca var. violacea, C. fusca var. flabellata, C. koreana, and C. brachyura. According to Johnson (1997) and Wang & Li (2005), these seven taxa belong to the sections Viorna (C. fusca var. fusca, C. fusca var. violacea, C. fusca var. flabellata), Atragene (C. ochotensis, C. calcicola, and C. koreana), and Pterocarpa (C. brachyura). In a recent phylogenetic classification of the genus, C. fusca var. fusca, C. fusca var. violacea, and C. fusca var. flabellata were classified under clade L; C. ochotensis, C. calcicola, and C. koreana were classified under clade H: and C. brachyura was classified under clade K (Lehtonen, Christenhusz & Falck, 2016). Additionally, C. urticifolia, C. takedana, and C. heracleifolia, which made up the third cluster in the UPGMA phenogram in this study, belong to section Tubulosae in the Johnson (1997) and Wang & Li (2005) classification and to clade C in Lehtonen, Christenhusz & Falck (2016). According to infrageneric classifications by Tamura (1995), Wang & Li (2005) and Xie, Wen & Li (2011), C. brachyura is considered to be closer to section Flammula (C. taeguensis, C. hexapetala, C. terniflora var. mandshurica, and C. terniflora). However, in this study, it remained connected to the Viorna and Atragene sections. In fact, this uncertainty is a common and typical interpretation for this large genus, as previous morphological and molecular studies also found a similar tendency (Grey-Wilson, 2000; Wang & Li, 2005; Miikeda et al., 2006; Xie, Wen & Li, 2011; Xie & Li, 2012; Lehtonen, Christenhusz & Falck, 2016; Ghimire et al., 2020).

According to PCA, the close affinity of the taxa from section Tubulosae is well retained by their proximity in axis 1 (Fig. 7). Previous studies have suggested that neither molecular analyses nor morphological data strongly support the infrageneric classification of Clematis. We agree with most authors that performing infrageneric classification of Clematis based on morphological characters is extremely difficult. In our analysis, out of the three taxa from section Viorna, C. fusca var. flabellata was grouped with C. koreana (section Atragene) and C. brachurya (section Pterocarpa), whereas C. fusca var. fusca and C. fusca var. violacea remained together in a separate sub-cluster. The PCA biplot also inferred similar relationships among these three subspecific taxa, with C. fusca var. flabellata was positioned on the positive side of axis 1, while C. fusca var. fusca and C. fusca var. violacea remained closer on the negative side of axis 1. The petioles of C. fusca var. flabellata differ from those of C. fusca var. fusca and C. fusca var. violacea in terms of the trichome abundance, phloem fibre cap height, and number of vascular bundles. In terms of morphology, C. fusca var. flabellata is an erect herb with ternate leaves, whereas C. fusca and C. fusca var. violacea are woody vines with pinnately foliate leaves. In addition, of the three species in section Clematis, C. trichotoma and C. brevicaudata formed a sub-cluster with C. patens (section Viticella), while C. apiifolia allied with C. terniflora (section Flammula) in another sub-cluster. The PCA showed a slightly different pattern, as C. apiifolia remained closer to C. serraatifolia and C. patens on the negative side of both axes, whereas C. brevicaudata allied with C. taeguensis and C. terniflora var. mandshurica on the positive side of axis 1 and the negative side of axis 2. Morphologically, the petiole of C. apiifolia differs from those of the other two species in its trichome abundance, outline in CS, phloem fibre cap, intrafascicular sclerenchyma, and number of vascular bundles. Notably,petiole features alone can be more useful for species delimitation than for distinguishing infrageneric relationships and thus are not indicative of infrageneric classifications. Similar explanations were provided by Ghimire et al. (2020) and Xie & Li (2012) based on the achene morphology and pollen morphology of Clematis, respectively.

In conclusion, the presence/absence and abundance of trichomes, petiole outline in CS, upper surface wings and groove, and number of vascular bundles were determined to be useful for species discrimination in Korean Clematis. The results of this study also indicated that taxa from the sections Tubulosae, Viorna, and Atragene could be grouped based on the overall similarity of their petiole features. Moreover, we found that many of the petiole characters such as the abundance of trichomes, upper surface wings and grooves, and number of vascular bundles, that were neglected in taxonomic and systematic considerations within and among Clematis taxa can be useful for species delimitation. We do not consider these petiole morphological data to be strong enough to provide useful information on infrageneric relationships with the Clematis genus; however, our results provide new and interesting insights into these data, which can be used as a source of descriptive and/or diagnostic features for particular taxa in the genus.

Supplemental Information

Supplemental Information 1 Character and character states of petiole of Clematis species

Click here for additional data file.

Additional Information and Declarations

Competing Interests

Author Contributions

Data Availability

The authors declare there are no competing interests.

Beom Kyun Park conceived and designed the experiments, performed the experiments, prepared figures and/or tables, authored or reviewed drafts of the paper, and approved the final draft.

Dong Chan Son conceived and designed the experiments, analyzed the data, authored or reviewed drafts of the paper, and approved the final draft.

Balkrishna Ghimire conceived and designed the experiments, performed the experiments, analyzed the data, prepared figures and/or tables, authored or reviewed drafts of the paper, and approved the final draft.

The following information was supplied regarding data availability:

All the raw data used in the statistical analysis are available in the Supplemental File.

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
