# Peer review of "Taxonomic relevance of petiole anatomical and micro-morphological characteristics of Clematis L. (Ranunculaceae) taxa from South Korea"

_PeerJ, doi:10.7717/peerj.11669_

## Round 0.1 · original submission · Major Revisions

Among crucial issues to be addressed and suggested by the reviewers are to discuss results based strictly on the interpretation of your petiole observations. This is perhaps the section that needs substantial improvements. Also, the figures should be enhanced. Revise citations and please cite previous work precisely. I agree with the concerns of the first reviewer in that the paper needs considerable work to reach the level of the papers published in PeerJ, please consider carefully the issues and suggestions raised by reviewer 1 in particular.

·

Basic reporting

BASIC REPORTING
The manuscript is poorly written and inaccurate in details. The authors often confuse terms ‘species’ and ‘taxa’; do not operate the Latin singular and plural forms, etc. At the beginning, they stated that their results should help in the taxonomy of the genus Clematis. However, in conclusions, they say that their interpretations (not the results! but exactly interpretations) can be arbitrary and, moreover, they recommend themselves to conduct more investigations. Such a position of the authors is at least a bit confusing. Many sentences in the manuscript are unrelated to the main topic. Many other sentences look like stumps - the authors just mention that somebody published something and that all - no any contextual discussion, nothing. Finally, the manuscript suffers from a lack of good and relative citations. The authors cite many old publications, many of which can be omitted, but do not mention really important and recent publications. For example, they declare that capitate glandular trichomes were never reported for petioles and leaves in the genus Clematis before. And this also confuse. Because, I am sure, many old publications report such trichomes (e.g., Svenson 1941; Keener 1967; Kapoor et al. 1985 and other). Many extensive citations, unclear structure of some sentences, and imperfect English make the manuscript difficult to read.

Experimental design

I believe that the study in general, methods, and outcomes are Ok, and lay in the journal field. The paper represents fascinating results and should be published after improving the manuscript and reconsidering its structure. In particular, it would be good to have a more in-depth discussion on the UPGMA tree. Because some of the close taxa (i.e., varieties of C. fusca) appeared far from each other. I also think that this manuscript can be reevaluated more precisely from the point of scientific soundness after improvement because, at the moment, it is difficult to read and understand.

Validity of the findings

This manuscript does not provide some super high-level investigations, but it represents interesting results that can be useful for further taxonomical explorations. However, it looks like the authors have difficulties with the interpretation of their results. Hence, consequently, conclusions are very unclear and controversial. Figures and tables look Ok. More of my comments and suggestions are provided in the attached file.

Reviewer 2 ·

Basic reporting

In my opinion, this paper is fairly good but I would strongly suggest a professional English proofread is needed. There are many grammar mistakes. And some sentence is in organized.

It got enough literature references and has sufficient field background. This article has been written In such a way where we can see the structure of this paper.

I would recommend the petiole morphology picture should be enhanced in order to get clear pictures, do some editing to those pictures using Adobe Photoshop,

Experimental design

This manuscript fulfills the aims and scope of this journal.

The findings answered all the research questions and the research structure is well defined, relevant, and meaningful for plant taxonomy research.

The study using adequate technical and method to fulfill its objective. And the methods were described with sufficient detail and informative.

All methods should be referred to previous used methods, citation is needed in this matter.

Validity of the findings

The results were accepted and well organized, however, I would strongly suggest the authors construct key identification of species to prove that the petiole morphological and anatomical characteristics definitely have taxonomic value for Clematis species studied. The rest is just fine.

Additional comments

Please amend all the mistakes that have been marked in the text. This manuscript has its own scientific value and definitely will be useful for plant systematic and taxonomic study.

Annotated reviews are not available for download in order to protect the identity of reviewers who chose to remain anonymous.

---

## Round 0.2 · Major Revisions

I agree with the reviewer whose recommendation was Major Revisions. Results of PCA should be better explained and discuss the groups retrieved in this analysis in a clear way. Check literature taking into account PeerJ format. In addition, consider the issues that this reviewer raised directly in the attached file.

·

Basic reporting

After the first revision, the manuscript has been significantly improved, so now it can be readable, which is a great success. However, minor spelling corrections and fixing the typos are still required, and I highlighted some of them in the attached file. The worse situation is with a representation of the results and some certain unclear moments in the manuscript. First of all, the manuscript is still overloaded with unnecessary citations with no strict relation to this study. For example, I have no clue why authors decided to cite “Illustrated encyclopedia of fauna & flora of Korea” and “Illustrated woody plants of Korea”, as well as how their outcomes are related to Apiacea, Lauracea, Dilleniaceae, Cunoniacae, and Piperales. Perhaps there is some unobvious logic, but then authors should be so kind to open it to not so sophisticated readers as I am. Moreover, in many places (indicated in the attached file), perhaps due to linguistic limitations, the authors wrote some unclear or controversial phrases that must be rewritten or explained before accepting this manuscript. Finally, the References are poorly formatted and lack DOIs, and must be re-checked. The other few comments I did directly in the text, so please follow them.

Experimental design

The experimental design is still fine. Nothing changed here after the first revision.

Validity of the findings

Findings are valid, just as before.

Additional comments

Dear authors, you did a great job, but there is still some work to do. Just keep in mind, we both want the same - make your publication better. Good luck!

Reviewer 2 ·

Basic reporting

I have read the manuscript and found out that this article has fairly good professional English used throughout the manuscript, with sufficient literature references, field background, structured figures, tables, and relevant results. There are only a few changes that need to be done to the manuscript.

Experimental design

Well defined but there are a few details in the referred methodology used in this study that need to be mentioned.

Validity of the findings

The findings are very well discussed and explained, and also clearly stated with figures helps. Conclusions are well stated that linked to the original objective of the study.

Additional comments

Please amend accordingly as stated in the text.

I would suggest the title to “Taxonomic relevance of petiole anatomical and morphological characteristics of Clematis L. (Ranunculaceae) taxa from South Korea’

Simplify the sentence, the sentence is a bit confusing in the abstract.

Add one more reference by V.H. Heywood, R. K. Brummitt A. Culham & O.Seberg (2019); Flowering Plant Families of The World.

Please specify how the identification of species has been done in this study.

Please specify the method used in this study referring to which references for stereo and microscopy technique.

Formalin is hazardous, is there any chance to used Acid acetic: Alcohol. Just curious.

Annotated reviews are not available for download in order to protect the identity of reviewers who chose to remain anonymous.

Reviewer 3 ·

Basic reporting

the English language is clear and professional.
as the study is concentrated on the anatomical and micromorphological analysis of petiole i think the title should be somehow changed as there is no morphological study. the paper has valuable data on less considered taxa but I think features as the depth of the groove is depended on the parts selected for the study. figures are of high quality and reasonably presented in the text.

Experimental design

the Experimental design is relevant. the samples and the method are ok. applied techniques are perfect. sufficient information is given about the process and methods used.
more populations should be used to extract such results but here one sample for each species is used.

Validity of the findings

I think the populations used for each species should be more than only one sample. if it is available the more replication should be added to these findings. what if the selected samples are not a good representative of the species?

Additional comments

due to the micromorphological and anatomical study, I think the title of the article should be revised. the electron micrographs of the petioles surface are very attractive but I think there should be more description of the observed variation.

---

## Round 0.3 · accepted · Accept

Thank you for considering all issues raised by the reviewers, they improved the article. I guess that Figures 2 and 7 will need to be improved, they seem one a little bit dark and the other too small. However, the editorial team will let you know whether they need changes.

·

Basic reporting

The authors did a good job! The first version of the manuscript was rather unreadable and very unclear in the representation of the outcomes. There are still some scientifically discursive moments in the manuscript, but its quality was significantly improved, and now it looks really interesting for reading and in-depth analysis. Hence, I believe that it should be published, and if somebody does similar research, he or she will be able to extend such discursive moments in its own paper - to criticize this work or, from the opposite, to support it with new findings. Anyway, now the manuscript looks like a strong finished and reliable scientific report and can be accepted for publication, I guess.

Experimental design

All is Ok and extensively explained in the previous revisions.

Validity of the findings

All is Ok and extensively explained in the previous revisions.

Additional comments

Many thanks to the authors for their work and patience in working on corrections!